# $H_\infty$ State Feedback Controller Based on Dynamic Observer Design for Singular Fractional-Order Systems

1st Minghui Wei
*Shenyang Aerospace University*
*School of Automation*
Shenyang, China
2271700918@qq.com

2nd He Li
*Shenyang Aerospace University*
*School of Automation*
Shenyang, China
lihe_good@126.com

3rd Shuo Liu
*Shenyang Aerospace University*
*School of Automation*
Shenyang, China
2922601793@qq.com

*Abstract*—This paper focuses on the problem of $H_\infty$ state feedback controller design based on dynamic observer for singular fractional-order systems (FOS), where the fractional derivative order $\alpha$ lies between 0 and 1. First, a new form of dynamic observer with a non-singular structure is proposed, which is easier to implement physically. Secondly, the bounded real lemma corresponding to $H_\infty$ norm of FOS is proposed via a set of linear matrix inequalities (LMIs). Compared to existing methods,the lemma employs real variable, which is easier to solve. Building upon the new lemma, the conditions for designing $H_\infty$ state feedback controller based on dynamic observer of FOS are derived. Finally, numerical example is presented to validate the effectiveness of the proposed method.

*Index Terms*—singular fractional-order systems (singular FOS), dynamic observer, $H_\infty$ control, state feedback control

## I. INTRODUCTION

In the past decade, fractional-order calculus has garnered considerable attention from physicists and engineers [1]. It has been observed that many systems across various interdisciplinary fields can be effectively described using fractional derivatives because these derivatives capture the historical evolution of functions and demonstrate stronger global correlations compared to integer derivatives. Numerous systems exhibit fractional-order dynamics, including viscoelastic systems [2], dielectric polarization [3], electrode-electrolyte polarization [4], electromagnetic waves [5], quantitative finance [6], and the quantum evolution of complex systems [7].

Singular systems, also known as generalized systems, encompass both differential and algebraic equations [8]. This model accounts for physical constraints, static relationships, and broader impulsive behaviors due to improper transfer matrices. In contrast to non-singular systems, singular FOS provide a more precise representation of the physical properties of systems, offering direct and comprehensive descriptions [9]. Since their introduction in many fields of system design and control, singular FOS have received considerable attention. They have diverse applications in electrical systems,

This study was funded by National Natural Science Foundation of China (grant number 62003223).

large-scale interconnected networks, power grids, constrained mechanical systems, and chemical processes [10]–[12].

In control system design, state feedback controllers are typically designed to meet specific performance criteria, especially when access to all states of the considered system is unavailable, or when system output measurements cannot provide complete information about the internal system states. This poses challenges for the design of state feedback controllers [13]. Therefore, the theory of observer design has attracted widespread attention. Based on state estimates obtained from the observer, the observer based controller is used to generate control laws to stabilize unstable systems or ensure desired performance [14], [15]. Recently, research activities focusing on observer-based control for FOS have been developed. In [22], a novel observer-free synchronization method is introduced for a specific category of incommensurate fractional-order systems. [23] studied the robust $H_\infty$ observer control of linear time-invariant perturbed uncertain FOS. By analyzing the $H_\infty$ norm of the FOS and considering the fractional derivative $\alpha$, a new sufficient condition is proposed to ensure the stability of the estimation error system.

$H_\infty$ control plays a crucial role in control systems. The $H_\infty$ optimization used under the presence of disturbances with bounded energy allows guaranteeing levels of disturbance attenuation. However, it is typically limited to integer-order systems [17]. In recent years, there have been developments extending the computation of $H_2$ and $H_\infty$ norms to FOS. The $H_2$ norm of fractional transfer functions of implicit type is studied in [16]. [18] employs two methods based on a binary algorithm and LMI condition to compute the $H_\infty$ norm of FOS and determine the Hamiltonian matrix. Using the generalized Kalman-Yakubovich-Popov (KYP) lemma, the bounded real lemmas for the $H_-$ norm and $H_\infty$ norm of FOS are derived via a series of LMIs in [19]. Based on these analysis results, numerous studies have focused on designing $H_\infty$ controllers and observers. [21] studied the finite time $H_\infty$ control problem of fractional order neural networks using finite time stability theory and Lyapunov sample function method. The $H_\infty$ control problem for singular FOS with order ranging

from 0 to 1 is explored in [20].

In this work, the problem of designing $H_\infty$ state feedback controller based on dynamic observers for singular FOS is studied. The main contributions can be summarized as follows:

- A dynamic observer is proposed. Compared with [26], the observer in this paper has a non-singular structure, making it easier to implement.
- Novel necessary and sufficient conditions for the bounded real lemma corresponding to $H_\infty$ norm for singular FOS ranging $0 < \alpha < 1$ are proposed. Unlike previous approaches, such as [24] and [25], the matrix variable is real, which is easier to solve.
- Based on the bounded real lemma, the conditions for designing the dynamic observer are given via a set of LMIs.

*Notations: In the subsequent sections of the paper, $A$ is a hermitian matrix if and only if $A^* = A$ and $A > 0$. $Re(Q)$ and $Im(Q)$ represent the real and imaginary parts of the complex $Q$, respectively. $Sym(A) = A + A^T$.*

**Proposition 1.** *A complex Hermitian matrix $Q$ satisfies $Q < 0$ if and only if*

$$\begin{bmatrix} R_e(Q) & I_m(Q) \\ -I_m(Q) & R_e(Q) \end{bmatrix} < 0$$

## II. PROBLEM STATEMENT AND PRELIMINARIES

Consider the following singular FOS:

$$\begin{cases} ED^\alpha x & = Ax(t) + Bu(t) + B_w w(t), \\ z(t) & = C_z x(t) + D_z u(t), \\ y(t) & = Cx(t), \end{cases} \tag{1}$$

in which $\alpha$ is the fractional-order, ranging from $0 < \alpha < 1$, $x \in R^n$ is the pseudo state vector, $y \in R^q$ is the output vector, $z \in R^r$ is the control output, $u \in R^m$ is the control input, $w \in R^p$ is disturbance input. $A$, $B$, $B_w$, $C$, $C_z$ are the constant matries for the appropriate dimensions, $E \in R^{n \times n}$ is the singular matrix, which is $rank(E) < n$. $D^\alpha$ denotes the Caputo fractional derivative

$$D^\alpha f(t) = \frac{1}{\Gamma(m-\alpha)} \int_{t_0}^t \frac{f^{(m)}(\tau)}{(t-\tau)^{\alpha+1-m}} d\tau. \tag{2}$$

In this paper, it is assumed that $E$ and $C$ such that

$$rank \begin{bmatrix} E \\ C \end{bmatrix} = n. \tag{3}$$

Then, consider the following observer based controller

$$\begin{cases} D^\alpha z(t) = & TA\hat{x}(t) + TBu(t) + C_d x_d(t) \\ & + D_d(y(t) - \hat{y}(t)), \\ D^\alpha x_d(t) = & A_d x_d(t) + B_d(y(t) - \hat{y}(t)), \\ \hat{x}(t) = & z(t) + Ny(t), \\ u(t) = & K\hat{x}(t), \end{cases} \tag{4}$$

in which $\hat{x}(t) \in R^n$ is the state estimation vector, $x_d(t) \in R^n$ is an auxiliary state, and $T$, $N$, $A_d$, $B_d$, $C_d$, $D_d$, $K$ are constant matrices of appropriate dimensions, and $T$, $N$ such that

$$TE + NC = I_n. \tag{5}$$

in which $I_n$ represents the n dimensional identity matrix.

Define $e(t) = x(t) - \hat{x}(t), \bar{x} = \begin{bmatrix} x^T & e^T & x_d^T \end{bmatrix}^T$. Combining singular FOS (1) and controller (2), one obtains

$$\begin{cases} \bar{E} D^\alpha \bar{x}(t) & = \bar{A}\bar{x}(t) + \bar{B}w(t), \\ z(t) & = \bar{C}\bar{x}(t), \end{cases} \tag{6}$$

where,

$$\bar{E} = \begin{bmatrix} E & 0 & 0 \\ 0 & I & 0 \\ 0 & 0 & I \end{bmatrix}, \bar{A} = \begin{bmatrix} A + BK & -BK & 0 \\ 0 & TA - D_d C & -C_d \\ 0 & B_d C & A_d \end{bmatrix},$$

$$\bar{B} = \begin{bmatrix} B_w \\ TB_w \\ 0 \end{bmatrix}, \bar{C} = \begin{bmatrix} C_z + D_z K & -D_z K & 0 \end{bmatrix}.$$

The transfer function of system (6) is

$$G(s) = \bar{C}(s^\alpha \bar{E} - \bar{A})^{-1} \bar{B}. \tag{7}$$

The design problem of $H_\infty$ state feedback controller based on dynamic observer is to design a controller such that the closed-loop system (6) is admissibility, and its transfer function satisfies $||G(s)||_\infty < \gamma$.

**Lemma 1.** *[27] Let $\gamma$ be a scalar such that $\gamma > 0$. The singular FOS is admissible and satisfies the condition $||G(s)||_\infty < \gamma$ if there exists $\bar{E}P = P^* \bar{E}^T \in \mathbf{C}^{n \times n} > 0$ such that*

$$\begin{bmatrix} Sym(\bar{A}(rP + \bar{r}\bar{P})) & * & * \\ \bar{C}(rP + \bar{r}\bar{P}) & -I & * \\ \bar{B}^T & 0 & -\gamma^2 I \end{bmatrix} < 0, \tag{8}$$

*where $r = e^{j\theta}$, $\theta = \frac{\pi}{2}(1 - \alpha)$.*

**Lemma 2.** *[28] Let $\gamma$ be a scalar such that $\gamma > 0$, the following statements hold the same significance:*

*(i) there exists $\bar{E}P = P^* \bar{E}^T \in \mathbf{C}^{n \times n} > 0$ such that*

$$\begin{bmatrix} Sym(\bar{A}(rP + \bar{r}\bar{P})) & * & * \\ \bar{C}(rP + \bar{r}\bar{P}) & -I & * \\ \bar{B}^T & 0 & -\gamma^2 I \end{bmatrix} < 0, \tag{9}$$

*where $r = e^{j\theta}$, $\theta = \frac{\pi}{2}(1 - \alpha)$.*

*(ii) there exists matrix $M \in \mathbf{R}^{n \times n}$ such that*

$$\begin{bmatrix} Sym(\bar{A}M) & * & * \\ \bar{C}M & -I & * \\ \bar{B}^T & 0 & -\gamma^2 I \end{bmatrix} < 0, \tag{10}$$

$$\begin{bmatrix} (\bar{E}M + (\bar{E}M)^T)/a & * \\ (\bar{E}M - (\bar{E}M)^T)/b & (\bar{E}M + \bar{E}M)^T)/a \end{bmatrix} > 0, \tag{11}$$

*where $\theta = \frac{\pi}{2}(1 - \alpha)$, $a = 4cos\theta$, $b = 4sin\theta$.*

*Proof.* Define $Q_1 = \bar{E}P_1 = R_e(Q)$, $Q_2 = \bar{E}P_2 = I_m(Q)$. According to Proposition 1, the condition $Q = \bar{E}P_1 + \bar{E}P_2 i > 0$ is equivalent to

$$\begin{bmatrix} \bar{E}P_1 & \bar{E}P_2 \\ -\bar{E}P_2 & \bar{E}P_1 \end{bmatrix} > 0. \tag{12}$$

Since $r = e^{j\theta} = cos\theta + isin\theta$ and $\bar{r} = e^{-j\theta} = cos\theta - isin\theta$, then it yields

$$
\begin{aligned}
(r\,Q + \bar{r}\bar{Q}) &= (cos\theta + sin\theta)(\bar{E}P_1 + \bar{E}P_2 i) \\
&\quad + (cos\theta - sin\theta)(\bar{E}P_1 - \bar{E}P_2 i) \quad (13) \\
&= (2cos\theta\bar{E}P_1 - 2sin\theta\bar{E}P_2).
\end{aligned}
$$

Note that $\bar{E}P_1$ is real symmetric matrix, while $\bar{E}P_2$ is skew-symmetric matrix. Therefore

$$
(r\,Q + \bar{r}\bar{Q})^T = (2cos\theta\bar{E}P_1 + 2sin\theta\bar{E}P_2). \quad (14)
$$

Let $\tilde{Q} = r\,Q + \bar{r}\bar{Q}$, $M = rP + \bar{r}\bar{P}$, then $\tilde{Q} = \bar{E}M$, Then (9) is equivalent to (10). From (13) and (14), we obtain

$$
\begin{cases}
\bar{E}P_1 = (\bar{E}M + (\bar{E}M)^T)/(4cos\theta) \\
\bar{E}P_2 = ((\bar{E}M)^T - \bar{E}M)/(4sin\theta).
\end{cases} \quad (15)
$$

Combining (15) and (12), it follows that (11) holds, which is equivalent to condition $Q = \bar{E}P_1 + \bar{E}P_2 i > 0$. This ends the proof. $\square$

## III. MAIN RESULTS

In this section, the problem of designing observer based controller is transformed into an optimization problem of LMIs. The conditions for designing $H_\infty$ state feedback controller based on dynamic observer for singular FOS are provided in the form of LMIs.

**Theorem 1.** *Let the system (6) be admissible and $\gamma > 0$. For $\|G(s)\|_\infty < \gamma$ if and only if there exists matrices $X$, $Y$, $H$, $S$, $G$, $L$ and $W$ such that*

$$
\begin{bmatrix}
\Omega_{11} & * & * & * & * \\
-(BS)^T & \Omega_{22} & * & * & * \\
A^T & \Omega_{32} & \Omega_{33} & * & * \\
\Omega_{41} & -D_z S & C_z & -\gamma I & * \\
B_w & B_w^T T^T & B_w^T T^T Y^T & 0 & -\gamma I
\end{bmatrix} < 0 \quad (16)
$$

$$
\begin{bmatrix}
(\mathbf{Q} + \mathbf{Q}^T)/a & * \\
(\mathbf{Q} - \mathbf{Q}^T)/b & (\mathbf{Q} + \mathbf{Q}^T)/a
\end{bmatrix} > 0, \quad (17)
$$

$$
\begin{aligned}
\Omega_{11} &= Sym(A\,X + BS), \\
\Omega_{22} &= Sym(TA\,X - H), \\
\Omega_{32} &= A^T T^T - C^T L^T + W, \\
\Omega_{33} &= Sym(YTA + GC), \\
\Omega_{41} &= C_z\,X + D_z S, \\
\mathbf{Q} &= \begin{bmatrix} EX & 0 & E \\ 0 & X & I \\ 0 & Z & Y \end{bmatrix}.
\end{aligned}
$$

*Define controller gain $K$ and $A_d$, $B_d$, $C_d$, $D_d$*

$$
\begin{aligned}
K &= S\,X^{-1}, \\
D_d &= L, \\
C_d &= (H - D_d CX)U^{-1}, \\
B_d &= V^{-1}(G + YD_d), \\
A_d &= V^{-1}(W - YTA\,X + Y\,D_d CX \\
&\quad - VB_d C\,X + Y\,C_d)U^{-1}, \quad (18)
\end{aligned}
$$

where $V$ and $U$ are invertible, which satisfies

$$
Z = Y\,X + VU. \quad (19)
$$

*Proof.* According to Lemma 2, the singular FOS is admissible and satisfies $\|G(s)\|_\infty < \gamma$ if there exists a matrix $M \in \mathbf{R}^{n \times n}$ such that

$$
\begin{bmatrix}
Sym(\bar{A}M) & * & * \\
\bar{C}M & -I & * \\
\bar{B}^T & 0 & -\gamma^2 I
\end{bmatrix} < 0, \quad (20)
$$

$$
\begin{bmatrix}
(\bar{E}M + (\bar{E}M)^T)/a & * \\
(\bar{E}M - (\bar{E}M)^T)/b & (\bar{E}M + \bar{E}M^T)/a
\end{bmatrix} > 0, \quad (21)
$$

Next, the target is to linearize the condition (20) as described in [29]. First, $M$ and $M^{-1}$ as

$$
M = \begin{bmatrix}
X & 0 & V^{-T} \\
0 & X & (I - XY^T)V^{-T} \\
0 & U & -UY^T V^{-T}
\end{bmatrix},
$$

$$
M^{-1} = \begin{bmatrix}
X^{-1} & -X^{-1} & U^{-1} \\
0 & Y^T & (I - Y^T X)U^{-1} \\
0 & V^T & -V^T X U^{-1}
\end{bmatrix},
$$

where $V$ and $U$ are invertible, which satisfies $Z = Y\,X + VU$, and define

$$
F = \begin{bmatrix}
I & 0 & 0 \\
0 & I & 0 \\
0 & Y & V
\end{bmatrix}.
$$

Multiplying both sides of (20) with the diagonal matrix $\{F, I, I\}$ and its transpose, we obtain

$$
\begin{bmatrix}
\tilde{\Omega}_{11} & * & * & * & * \\
-(BKX)^T & \tilde{\Omega}_{22} & * & * & * \\
A^T & \tilde{\Omega}_{32} & \tilde{\Omega}_{33} & * & * \\
\tilde{\Omega}_{41} & -D_z KX & C_z & -\gamma I & * \\
B_w^T & B_w^T T^T & B_w^T T^T Y^T & 0 & -\gamma I
\end{bmatrix} < 0 \quad (22)
$$

where

$$
\begin{aligned}
\tilde{\Omega}_{11} &= Sym(A\,X + BKX), \\
\tilde{\Omega}_{22} &= Sym(TA\,X - D_d C\,X - C_d U), \\
\tilde{\Omega}_{32} &= YTA\,X - Y\,D_d C\,X + VB_d C\,X - Y\,C_d U \\
&\quad + V A_d U + (TA - D_d C)^T, \quad (23) \\
\tilde{\Omega}_{33} &= Sym(YTA - YD_d C + VB_d C), \\
\tilde{\Omega}_{41} &= C_z\,X + D_z KX\,.
\end{aligned}
$$

Let

$$
\begin{aligned}
S &= KX, \quad (24) \\
H &= D_d C\,X + C_d U, \\
W &= YTA\,X - Y\,D_d C\,X + VB_d CX \\
&\quad - YC_d U + V A_d U, \\
G &= VB_d - YD_d.
\end{aligned}
$$

Combining (24) and (22), it follows that (16) holds.

Then multiplying both sides of (21) with the $F$ and $F^T$, and define $Z = Y\,X + VU$, this concludes the proof. $\square$

## IV. NUMERICAL EXAMPLE

Consider the fractional-order circuit depicted in Fig.1 [30], which can be characterized as

$$e_1 = L_1 \frac{d^\alpha i_1}{dt^\alpha} + L_3 \frac{d^\alpha i_3}{dt^\alpha} + R_1 i_1 + R_3 i_3,$$

$$e_2 = L_2 \frac{d^\alpha i_1}{dt^\alpha} + L_3 \frac{d^\alpha i_3}{dt^\alpha} + R_2 i_2 + R_3 i_3,$$

$$i_3 = i_1 + i_2.$$

Let $x(t) = \begin{bmatrix} i_1 \\ i_2 \\ i_3 \end{bmatrix}, z(t) = Cx(t)$. Then

$$\begin{bmatrix} L_1 & 0 & L_3 \\ 0 & L_2 & L_3 \\ 0 & 0 & 0 \end{bmatrix} D^\alpha x(t) = \begin{bmatrix} -R_1 & 0 & -R_3 \\ 0 & -R_2 & -R_3 \\ 1 & 1 & -1 \end{bmatrix} x(t)$$

$$+ \begin{bmatrix} 1 & 0 \\ 0 & 1 \\ 0 & 0 \end{bmatrix} u(t) + \begin{bmatrix} 1 \\ 1 \\ 1 \end{bmatrix} w,$$

$$z(t) = \begin{bmatrix} 1 & 1 & 1 \end{bmatrix} x(t).$$

Consider the appropriate parameters as follows

$$A = \begin{bmatrix} -4 & 0 & -5 \\ 0 & -3 & -5 \\ 1 & 1 & -1 \end{bmatrix}, B = \begin{bmatrix} 1 & 0 \\ 0 & 1 \\ 0 & 0 \end{bmatrix}, B_w = \begin{bmatrix} 1 \\ 1 \\ 1 \end{bmatrix},$$

$$C = \begin{bmatrix} 0 & 1 & 0 \\ 0 & 0 & 1 \end{bmatrix}, C_z = \begin{bmatrix} 1 & 1 & 1 \end{bmatrix}, D_z = \begin{bmatrix} 1 & 0 \end{bmatrix},$$

$$E = \begin{bmatrix} 1 & 0 & 0 \\ 0 & 1 & 0 \\ 0 & 0 & 0 \end{bmatrix}, \alpha = 1/3. \tag{25}$$

Then the solution to (5) is

$$T = \begin{bmatrix} 1 & -1 & 2 \\ 0 & 1 & 3 \\ 0 & 1 & 3 \end{bmatrix}, N = \begin{bmatrix} 1 & 0 \\ 0 & 0 \\ -1 & 1 \end{bmatrix}.$$

According to Theorem 1, we obtain that

$$A_d = \begin{bmatrix} -0.0540 & 0.8565 & -0.0369 \\ -0.6895 & -0.8754 & -0.6877 \\ 3.0035 & 3.2893 & 0.8686 \end{bmatrix},$$

$$B_d = \begin{bmatrix} 0.0422 & 0.0644 \\ 0.9692 & -0.1048 \\ -1.9431 & -0.8367 \end{bmatrix},$$

$$C_d = \begin{bmatrix} -0.1496 & 2.9450 & -0.0947 \\ 2.7666 & 2.0096 & -1.9358 \\ -0.5109 & -2.9046 & -2.1334 \end{bmatrix},$$

$$D_d = \begin{bmatrix} -0.0288 & -0.0662 \\ 0.3001 & 0.3680 \\ 0.7859 & 1.0324 \end{bmatrix},$$

$$K = \begin{bmatrix} 0.3793 & -0.2046 & 0.1052 \\ -0.3196 & 0.6161 & 1.4726 \end{bmatrix}.$$

Fig.2 shows state vector and state estimation vector. The maximum singular values of $G(s)$ are plotted in Fig. 3, peaking at approximately 0.4031. The state diagram of system (6) illustrated in Fig.4. It implies that the system is stable and admissible.

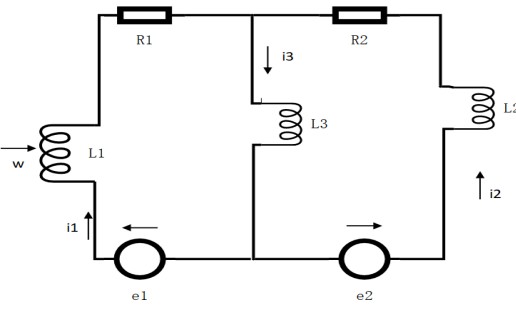

Fig. 1. The singular fractional-order electrical circuit.

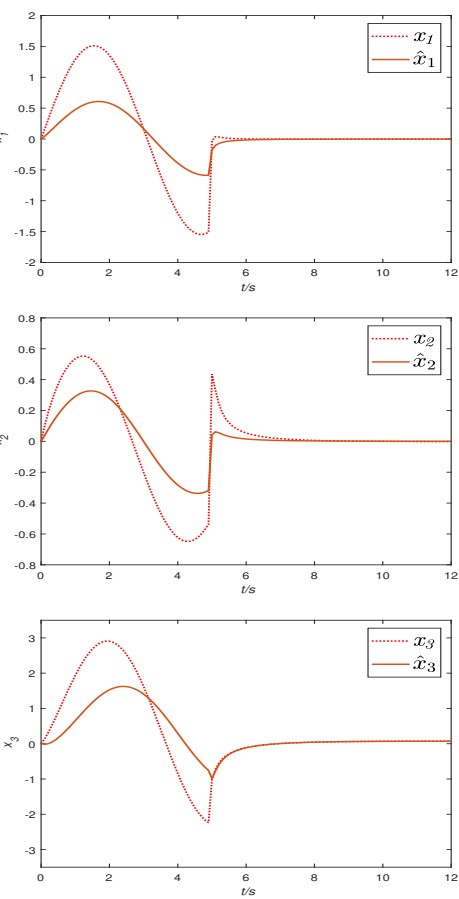

Fig. 2. State vector and state estimation vector.

## V. CONCLUSION

The problem of designing $H_\infty$ state feedback controller based on dynamic observer for singular FOS is investigated in this paper. Firstly, a non-singular $H_\infty$ state feedback controller based on dynamic observer is proposed. Additionally, a new

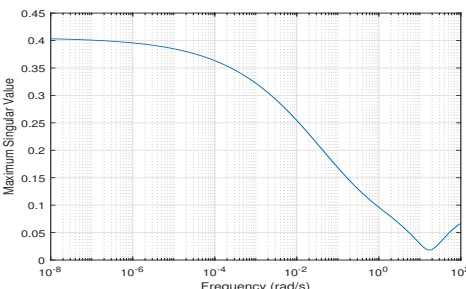

Fig. 3. The maximum singular values of $G(s)$.

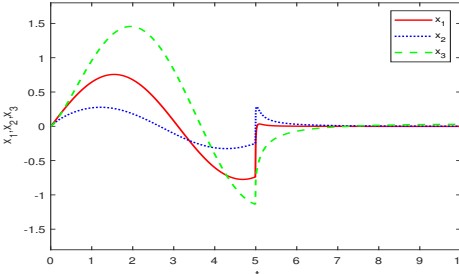

Fig. 4. State diagram of system (6).

bounded real lemma for the $H_\infty$ norm of FOS is presented, which forms the foundation for designing the dynamic observer. Then, conditions for designing $H_\infty$ state feedback controller based on dynamic observer for FOS are derived. Ultimately, the effectiveness of the proposed methodology is confirmed through a simulation example.

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
