# OpenReview forum: "H∞ State Feedback Controller Based on Dynamic Observer Design for Singular Fractional-Order Systems"
_IEEE.org/ICIST/2024/Conference — IEEE ICIST 2024 Conference Submission_

### Official Review · Reviewer_khe4 · 2024-08-29
**This paper can be accepted.**

**Rating:** 7
**Confidence:** 4

**Review:**

This paper is well written and the contributions are solid. However, I have some suggestions
1. The definitions of some notations should be given, for example, $R^{r}$, $\mathbf C$.
2. The references should be updated. For example, some dynamic controller design results,  IEEE TFS 26 (6), 3301-3313.
3. The motivation of the study of fractional-order systems should be strengthened.

---

### Official Review · Reviewer_pRc8 · 2024-08-30
**This paper can be accepted.**

**Rating:** 6
**Confidence:** 5

**Review:**

1. While the paper presents a solid approach to the design of H∞ state feedback controllers for singular fractional-order systems (FOS), the level of innovation could be strengthened. It would be helpful to clarify how the proposed method significantly advances or differs from existing approaches in the literature. What specific advantages does the new dynamic observer structure offer over traditional methods beyond ease of physical implementation?
2. It was noted that the formatting of the references is not consistent throughout the paper. Please ensure that all references adhere to a uniform citation style, as this is important for the overall professionalism and readability of the paper.

---

### Decision · Program_Chairs · 2024-09-06

Accept (Oral)